# Transcutaneous Laryngeal Ultrasound for Vocal Cord Paralysis Assessment in Patients Undergoing Thyroid and Parathyroid Surgery—A Systematic Review and Meta-Analysis

**DOI:** 10.3390/jcm10225393

**Published:** 2021-11-19

**Authors:** Agastya Patel, Piotr Spychalski, Aleksander Aszkiełowicz, Bogusław Mikaszewski, Jarek Kobiela

**Affiliations:** 1Department of General, Endocrine and Transplant Surgery, Medical University of Gdansk, 80-214 Gdansk, Poland; piotr.spychalski@gumed.edu.pl (P.S.); kobiela@gumed.edu.pl (J.K.); 2First Doctoral School, Medical University of Gdansk, 80-211 Gdansk, Poland; 3Department of Anesthesiology and Intensive Care, Medical University of Gdansk, 80-214 Gdansk, Poland; aleksander.aszkielowicz@gumed.edu.pl; 4Department of Otolaryngology, Medical University of Gdansk, 80-214 Gdansk, Poland; boguslaw.mikaszewski@gumed.edu.pl

**Keywords:** transcutaneous laryngeal ultrasonography, laryngoscopy, vocal cord paralysis, thyroidectomy, diagnostic accuracy

## Abstract

Recurrent laryngeal nerve injury is an important complication following thyroid and parathyroid surgery. Recently, Transcutaneous laryngeal ultrasound (TLUSG) has emerged as a non-invasive alternative to laryngoscopic examination for vocal cord (VC) assessment. The aim of the systematic review and meta-analysis was to determine its diagnostic accuracy in reference to laryngoscopy. It was conducted in accordance with PRISMA (Preferred Reporting Items for Systematic Reviews and Meta-Analyses) guidelines. MEDLINE, Scopus, Cochrane library and Web of Science databases were searched to identify relevant articles. Sixteen studies were included in the review. Pooled diagnostic accuracy was calculated based on weighted arithmetic mean and plotting forest plot. The pooled visualization rate was 86.28% and 94.13% preoperatively and postoperatively, respectively. The respective pooled sensitivity and specificity was 78.48% and 98.28%, and 83.96% (CI 95%: 77.24–88.50%) and 96.15% (CI 95%: 95.24–96.88%). The diagnostic accuracy improved if transverse and lateral approaches, and valsalva maneuver were utilized. Male gender and older age were the most crucial risk factors for VC non-visualization. TLUSG is an efficacious screening tool for vocal cord palsy due to its high sensitivity. It is likely to prevent unnecessary laryngoscopic examination in around 80% of patients, with the potential for becoming a gold standard for specific (female/young) patient cohort through assimilative modifications use, increasing expertise and development of objective measurements in the future.

## 1. Introduction

Thyroidectomy and parathyroidectomy are commonly performed surgical procedures for the treatment of benign and malignant diseases of these glands [1,2,3]. Recurrent laryngeal nerve (RLN) injury is an important complication following these procedures and may result in either temporary or permanent vocal cord palsy (VCP). This complication has important implications not only for the patient but also for the surgeon. For patients, VCP may either remain asymptomatic, or present with transient vocal fatigue or profound dysphonia and dyspnea, which substantially affects quality of life and recovery. For surgeons, it has been identified as a major cause for medicolegal litigation following thyroid procedures [4].

A 2009 systematic review reported that the incidence of transient and permanent VCP ranged from 1.4 to 38.4% and from 0 to 18.6%, respectively [5]. Several factors influence the rate of postoperative VCP: type and extent of surgical resection, underlying thyroid disease, intraoperative RLN identification and neuromonitoring, and surgeon’s experience [6,7]. The diagnostic approach used to identify VCP also influences the reported incidence rates. Several modalities are available for assessing VCP such as indirect laryngoscopy, direct flexible laryngoscopy (DFL), videostrobolaryngoscopy (VL), and the relatively novel, transcutaneous laryngeal ultrasonography (TLUSG) [8].

The assessment of vocal cords (VC) in the preoperative and postoperative period is crucial. In the preoperative period, it helps establish baseline characteristics and identify pre-existing VCP, whereas, postoperatively, early identification of VCP helps institute rapid management plan. There is an ongoing debate as to whether VC assessment should be performed routinely in either period. Current guidelines recommend preoperative laryngeal examination only for patients with impaired voice (based on self-report or evaluation scales), or with “nerves-at-risk” (known VCP, prior neck or mediastinal surgery, locally advanced malignancy, or large goiter), while postoperative assessment is recommended for patients only with voice changes [8,9].

The aforementioned modalities, including TLUSG, have been recommended as potential tools for laryngeal examination [8]. However, they have varying diagnostic accuracy, with DFL still considered as the “reference” standard for VC examination [5]. The primary advantage of DFL is the ability to visualize VC in over 99% cases [10]. Nonetheless, it is an invasive procedure, which may be painful and uncomfortable for the patients. It often necessitates a referral to an otolaryngologist, increasing healthcare cost and management time, unless the operating surgeon is experienced in laryngoscopy.

Over the last decade, TLUSG has emerged as an alternative modality for assessing VC mobility. It is a non-invasive procedure easily performed in an out-patient setting, while being cost-effective, time-efficient, less painful and easily tolerable [11]. During the ongoing COVID-19 pandemic, TLUSG has been noted as a safer alternative due to its non-aerosol producing nature [12]. However, there are concerns regarding its ability to accurately visualize and assess VC movement. Da costa et al., in their systematic review, suggested that TLUSG is a viable screening tool for laryngeal assessment post-thyroidectomy [11]. However, to date, a diagnostic test accuracy-based systematic review and meta-analysis comparing TLUSG to traditional modalities such as DFL and VL is missing.

The aim of the systematic review and meta-analysis is to assess the diagnostic accuracy of TLSUG with DFL or VL as the reference standard in the pre- and post- surgery setting. The review also aimed at identifying risk factors for non-visualization of VC on TLUSG examination and for determining the impact of technical variants on diagnostic accuracy of TLUSG.

## 2. Materials and Methods

### 2.1. Search Strategy

The systematic review and meta-analysis was performed based on a PICO format (patients, interventions, comparisons, outcomes) research question in accordance with the PRISMA (Preferred Reporting Items for Systematic Reviews and Meta-Analyses) statement. The following electronic databases were searched to identify articles: MEDLINE, Scopus, Cochrane library and Web of Science, without using any filters. Backward chaining of all suitable full-text articles was additionally performed to identify relevant articles.

### 2.2. Evidence Acquisition

On the 22 June 2021, two independent researchers (A.P., P.S.) performed a search of the MEDLINE, Scopus, Cochrane library and Web of Science databases to identify studies eligible for the review. The search strategy utilized was as follows: (“laryngeal ultrasonography” OR “TLUSG” OR “TLUS” OR “laryngeal ultrasound” OR “transcutaneous ultrasonography” OR “transcutaneous laryngeal ultrasonography”) AND (“thyroidectomy” OR “thyroid surgery” OR “vocal cord” OR “vocal cord palsy” OR “mobility” OR “parathyroid surgery”). A total of 166 articles were identified upon initial search. The PRISMA protocol is presented as Appendix A. The abstracts of these articles were screened by two independent researchers (A.P., P.S.). Any disagreements were resolved by arbitration in consultation with a third researcher (J.K.). The following information for each individual study was retrieved: first author, year of publication, country of study center, accrual years, exclusion criteria, number of patients, percentage of female patients, mean age, details on TLUSG and laryngoscopic examination, technical information related to TLUSG (probe type and frequency, approach, patient position, maneuvers used and criteria for diagnosing VCP) and factors influencing visualization rate. Various technical aspects of TLUSG are defined in Appendix A. Additionally, data on outcome measures, including visualization rate, sensitivity, specificity, positive predictive value (PPV), negative predictive value (NPV), and VCP rate reported in the articles were abstracted.

### 2.3. Inclusion and Exclusion Criteria

A PICO framework question was developed to identify relevant studies for inclusion in the review: Patients—patients undergoing thyroidectomy/parathyroidectomy for benign or malignant diseases; Intervention—VC assessment using TLUSG; Comparisons—VC assessment using DFL or VL; Outcomes—visualization rate, sensitivity, specificity, PPV, NPV and risk factors for non-visualization. Only full-text prospective studies comparing TLUSG with DFL or VL, in the setting of thyroidectomy or parathyroidectomy were included. Articles were excluded if TLUSG examination was not in the setting of thyroidectomy/parathyroidectomy, or lacked comparison with DFL/VL; if studies included patients with previous cervical surgery, were retrospective analyses, studied TLUSG learning curve or subjective assessment methods, did not assess VCP, did not report required outcome measures, and if articles were not original studies in English.

### 2.4. Statistical Analysis and Evidence Synthesis

The included studies were dichotomized into categories based on the reporting of diagnostic accuracy of TLUSG: (A) in terms of preoperative and postoperative assessment and (B) in terms of technical aspects of the examination (such as probe frequency, approach, maneuvers, sonographic landmarks, subjective assessment).

Indices of diagnostic accuracy were extracted from each included study for TLUSG and DFL or VL. Contingency tables in a 2 × 2 format were constructed for true positives, true negatives, false positives, and false negatives. For one study, such a table could not be prepared and reported sensitivity and specificity were retrieved directly and used for analyses.

VCP was defined as decreased or absent VC movement on laryngoscopic examination. Cases were defined as true positive and true negative when findings of VCP or normal VC movement on TLUSG was corroborated by DFL/VL, respectively. False positive cases were defined as patients with signs of VCP on TLUSG showing normal VC movement on DFL/VL, while false negative cases were defined as patients with normal VC movement on TLUSG showing VCP on DFL/VL. Sensitivity, specificity, PPV and NPV with a 95% confidence interval (CI) were calculated through Fisher’s exact test using GraphPad Prism 8.0 software (GraphPad Software Inc., San Diego, CA, USA).

Weighted arithmetic means (WAM) for visualization rate, VCP rate, sensitivity, specificity, PPV and NPV was calculated to establish pooled estimates for these measures. For visualization rate and VCP rate, WAM was calculated based on the total number of patients in each study. Diagnostic accuracy measurements were calculated based on the number of patients whose VC were visualized on TLUSG examination. If multiple papers reported on the same cohort, only results from the latest publication were considered for a particular analysis. Additionally, one study was excluded from the quantitative analysis as its diagnostic accuracy calculations were based on number of VC instead of patients [13]. Such calculations are liable to under- or over-estimation of diagnostic accuracy indices, as noted by extreme outlier results of this study compared to other included studies. The study also lacked data on visualization rates making it difficult to determine whether diagnostic accuracy calculations considered only patients with visualized VC. The remaining studies determined diagnostic accuracy based on patients in whom VC were visualized and were eligible for quantitative analysis.

Forest plots were created for sensitivity and specificity of TLUSG in the postoperative setting using RevMan 5.4 software (Cochrane, London, UK). Subgroup analyses were performed to assess accuracy of TLUSG based on two technical aspects (examination approach and maneuvers used) as well as based on the specialty of TLUSG examiner.

The methodological quality of the included studies was evaluated by using the Quality Assessment of Diagnostic Accuracy Studies-2 (QUADAS-2) tool. It includes four domains: patient selection, index test, reference standard, and flow and timing. Definitions and judgment criteria for each domain have been described elsewhere [14].

## 3. Results

Sixteen prospective studies were included in the review following abstract and full-text screening. Eleven studies reported on diagnostic accuracy of TLUSG in either preoperative period (*n* = 1), postoperative setting (*n* = 7), or both (*n* = 3) [14,15,16,17,18,19,20,21,22,23,24]. All of these studies, except one, were included in quantitative analysis to determine diagnostic accuracy of TLUSG in preoperative and postoperative period [13], while the remaining five studies focusing on specific technical aspects of TLUSG were reported in a descriptive manner to elucidate the influence of various technical aspects on TLUSG’s diagnostic accuracy [25,26,27,28,29].

### 3.1. Operative Period

A total of 3332 patients, across 10 studies, undergoing either thyroidectomy or parathyroidectomy procedures were included in the quantitative analysis. The percentage of female patients ranged from 66.2 to 87%, while the mean age ranged from 42 to 56 years.

Eight studies compared the diagnostic accuracy of TLUSG with DFL [15,17,18,19,20,22,23,24], while the remaining two studies compared it with VL [16,21]. The study protocols and the examiner experience varied across studies. The main characteristics of included studies are presented in Table 1. The additional details regarding patient selection criteria, TLUSG examination, study protocol and examiner experience are described in Appendix A.

#### 3.1.1. Diagnostic Accuracy Assessment

The weighted arithmetic mean of diagnostic accuracy indices, visualization rate and VCP rates are presented in Table 2.

Three studies assessed the diagnostic accuracy of TLUSG in the preoperative period [16,17,19]. The WAM for VC visualization rate in the preoperative period was 86.28% (range: 79–96.5%). The rate of VCP was 3.57% (range: 0.45–7.8%). The WAM for sensitivity, specificity, PPV and NPV was 78.48%, 98.28%, 86.42% and 99.73%, respectively. The summary of diagnostic accuracy statistics for TLUSG assessment in the preoperative period is provided in Appendix A. The sensitivity of TLUSG was greater in studies performing a transverse and lateral approach, and in those performing valsalva maneuver in comparison to those performing only transverse approach and those not utilizing valsalva maneuver in the preoperative period, respectively. (Table 2).

Nine included studies assessed diagnostic accuracy of TLUSG in the postoperative period [15,16,18,19,20,21,22,23,24]. Of these, three studies reported on the same cohort, therefore, only the most recent data for a particular analysis was considered [16,23,24]. The WAM for VC visualization rate in the postoperative period was 94.13% (range: 72.8–100%). The rate of VCP was 7.11% (range: 5.4–16.1%). The WAM for sensitivity, specificity, PPV and NPV was 83.95% (95% CI: 77.24–88.50%), 96.15% (95% CI: 95.24–96.88%), 64.32% (95% CI: 55.90–68.56%) and 98.71% (95% CI: 98.13–99.11%), respectively. The summary of diagnostic accuracy statistics for TLUSG assessment in the postoperative period is provided in Appendix A. The forest plot for pooled sensitivity and specificity of TLUSG in postoperative period is presented as Figure 1.

Four studies utilized the lateral approach in addition to the transverse approach [16,17,18,20], while only four studies performed the transverse approach [15,19,21,24]. The VC visualization rate was slightly higher in the studies using a combined approach (transverse and lateral, WAM = 96.04%) than those using only transverse approach (WAM = 93.11%). Similarly, all diagnostic accuracy indices were higher for studies using a combined approach than for those using only a transverse approach (Table 2).

In terms of maneuvers, particularly the valsalva maneuver was performed in two studies [19,24] and not performed in four studies [15,16,18,23]. Two studies did not perform any maneuvers as TLUSG examination was performed in the immediate postoperative period [20,21]. The WAM of VC visualization rate was 93.23% in studies not performing valsalva maneuver. Only one study performing valsalva maneuver in the postoperative period reported a visualization rate of 94.6% [24]. The sensitivity and NPV for TLUSG was higher in studies performing valsalva maneuver than those not performing it (Table 2).

On subgroup analysis based on examiner specialty, VC visualization rate was found to be comparable regardless of whether the operating surgeon, anesthesiologist or radiologist performed TLUSG. (Table 2). The diagnostic accuracy of TLUSG was greater in studies in which the operating surgeon or anesthesiologist performed TLUSG compared those with radiologists. (Table 2) This finding should be considered along with the reported experience of TLUSG examiners, which was considerably greater for the operating surgeons than the anesthesiologists or radiologists (Appendix A).

#### 3.1.2. Risk Factors for Vocal Cord Non-Visualization

The factors associated with non-visualization of VC during TLUSG examination were examined in seven included studies [13,15,17,18,19,22,23]. The majority of the studies identified male gender and older patient’s age at examination as significant risk factors for non-visualization of VC with TLUSG. (Table 3) Other factors negatively impacting visualization rates were body mass index, patient’s weight and height, anatomical features (distance between hyoid bone to cricoid cartilage to sternal notch), thyroid volume, operative time, and use of postoperative drainage. It should be noted that majority of the studies assessing the effect of BMI (four out of five) found that it did not affect VC visualization [15,17,18,23]. Underlying disease, type of surgery, thyroid function, use of cautery, and reoperation were factors that did not affect VC visualization.

### 3.2. Technical Modifications

Five prospective studies focused on individual technical aspects of TLUSG such as the effect of different maneuvers, using gel pad, USG frequency, different approaches and number of landmarks visualized on the visualization rate and diagnostic accuracy indices [25,26,27,28,29]. All of these studies compared TLUSG with the reference standard of DFL, with examiners of both tests blinded to each other. The characteristics of these studies are presented in Appendix A. The summary of visualization rates and diagnostic accuracy indices for these studies are presented in Appendix A.

Wong et al. compared three maneuvers (passive, active and valsalva) performed during TLUSG examination, in terms of their effect on VC assessment [25]. The authors found that performing valsalva maneuver resulted in higher specificity than passive maneuver (97.3% vs. 94.3%, *p* = 0.015) and greater VC visualization rate than with passive or active maneuver (92.1%, 91.5%, 89.8%, respectively).

Woo et al. assessed whether using gel pads improved the USG probe contact and wave penetration in comparison to traditional TLUSG method (no gel pad) [26]. The use of gel pads significantly improved visualization rates of VC landmarks compared to the traditional method (99% vs. 93.4%, *p* < 0.001). The diagnostic accuracy indices were similar between the two methods.

The same research group assessed the influence of a high frequency (5–12 MHz) versus a low frequency (3–9 MHz) probe for TLUSG examination [27]. Low frequency probe use resulted in a higher VC visualization rate than high frequency probe (97.7% vs. 88.4%, *p* < 0.001). The diagnostic accuracy of TLUSG remained similar regardless of USG frequency.

Woo and colleagues also assessed the performance of TLUSG using transverse approach for female patients and lateral approach for male patients [28]. The lateral approach in male patients resulted in same VC visualization rates compared to transverse approach in female patients (100% vs. 100%). The indices of diagnostic accuracy remained comparable between the two groups.

Wong et al. studied the importance of different landmarks identified during TLUSG examination and the technique’s diagnostic performance [29]. The study individually assessed true VC, false VC and arytenoid folds for this purpose. The authors concluded that identification of all three structures did not improve VCP assessment through TLUSG. Hence, it was not necessary to assess all three structures, rather it was enough to visualize normal movement in any one of the structures to exclude VCP.

### 3.3. Risk of Bias Assessment

The risk of bias and applicability concerns for included studies are shown in Appendix A. Overall, all included studies were judged as having low risk of bias. All expect two studies met at least four out of seven items of the QUADAS-2 tool. The remaining two studies met three of the seven items. Flow and timing domain may have introduced bias, with 8 studies judged as high risk, primarily due to not all patients receiving the reference or index test.

## 4. Discussion

This systematic review and meta-analysis reaffirm that VC visualization is more difficult and inferior with TLUSG examination than DLF. It also provides evidence that TLUSG is an ideal screening test for VCP due to its high sensitivity and high NPV. Several variants in the technique of TLUSG examinations have been proposed to improve its diagnostic performance. This review identifies that the use of lateral approach, valsalva maneuver, gel pads and low frequency USG probes enhances the diagnostic accuracy of TLUSG, and potentially VC visualization. An unexpected finding was that the VC visualization rates, and sensitivity were lower in the preoperative period compared to the postoperative period. However, this could simply be the consequence of deficiency in studies appraising TLUSG in the preoperative period.

As mentioned, the VC visualization during TLUSG examination remains suboptimal as it is achieved in about 86–94% patients compared to over 99% with DFL. The principal mechanism of TLUSG is the propagation of USG waves from the skin of the neck through the thyroid cartilage to the VC and back to the probe [30]. Two demographic characteristics are highlighted as crucial factors predisposing to VC non-visualization—older age and male gender. The thyroid cartilage undergoes progressive calcification with age, while its angle is much sharper in males than in females [31,32]. These factors reduce USG wave propagation resulting in lower image quality and increased difficulty in visualizing VC. To tackle this issue, several technical modifications have been proposed. The lateral approach involves placing the USG probe on a thinner and smoother lamina of the thyroid cartilage, which improves skin-to-probe contact and reduces distance from internal laryngeal structures. With valsalva maneuver, the VC adduct to the midline, allowing for easier visualization and mobility assessment. Low frequency USG have longer wavelengths and are less easily absorbed, which augments wave penetration through calcified thyroid cartilage, thereby improving image quality. Lastly, gel pads also improve skin-to-probe contact and enhance wave propagation. Consequently, our analysis demonstrated the advantage of utilizing a lateral approach and valsalva maneuver in enhancing the efficacy of TLUSG. A recent report highlighted the role of gel pads in dramatically improving VC visualization in male patients from 35% with traditional TLUSG to 78% with gel pad assisted TLUSG (*p* < 0.001) [33]. Similar enhancement was seen in the older patients (>50 years), in whom gel pad use increased visualization rate from 25% to 45% (*p* < 0.05). Therefore, through summation of these techniques, future technological advancements and appropriate patient selection, the visualization of VC landmarks through TLUSG can be effectively improved.

TLUSG is found to be a highly sensitive test for VCP indicating that it can effectively identify patients with the condition (i.e., high true positive, low false negative rate). Based on our analysis, TLUSG may fail to visualize VC in about six out of every 100 individuals postoperatively, and correctly identify VCP in 79 of the remaining 94 individuals. This translates into approximately 80% of patients being saved from unnecessary invasive DFL testing. Additionally, TLUSG appears to have a high NPV, further emphasizing its merit as a screening test. However, predictive values are dependent on the prevalence of the condition (i.e., VCP). Therefore, it is possible that the high NPV and low PPV seen in our analysis is due to the low pooled prevalence of VCP (7.11% postoperatively).

On the other hand, the accuracy of TLUSG is largely dependent on the subjective assessment of the examiner. The experience of examiners in the included studies varied considerably, ranging from either no prior experience or training in a single course to over 500 TLUSG examinations per year [13,15,16,19]. In most studies, a single examiner performed all TLUSG assessments bringing into question the generalizability of the results. Kandil et al. compared the accuracy of TLUSG results in the first and last 125 patients of their cohort, and found no improvement in outcomes with increasing experience [13]. To the contrary, Wong et al. on their assessment of learning curve of eight surgical residents, each performing 80 TLUSG examinations, reported that adequate mastery in TLUSG can be achieved after 40 examinations [34]. Their conclusion was based on the reduction in time duration for completing the examination. However, the diagnostic accuracy and visualization rates remained similar throughout the study period. Based on current evidence, it cannot be ascertained if TLUSG is an easily learned modality with reproduceable results as its diagnostic accuracy does not improve with increasing experience and the majority of the results are based on single-examiner assessment. There is a need for prospective evaluation of the learning curve for TLUSG, identifying factors influencing training, and determining the appropriate medical specialist to undertake such examination and if outcomes can be standardized amongst peers. Furthermore, methods for quantitative evaluation of VC mobility need to be developed and refined to introduce objectiveness in TLUSG assessment. To date, three quantitative methods have been described: symmetry index, mobility index and VC displacement velocity [35,36,37]. However, there is a lack of data, standardization and congruency between their outcomes and effectiveness.

In addition to RLN injury, voice abnormalities after thyroidectomy may occur due to the injury to the external branch of superior laryngeal nerve (EBSLN) [38]. The incidence of EBSLN injury remains unclear, especially considering the myriad of factors that may affect vocal cord function [38]. EBSLN innervates the cricothyroid muscle, with its dysfunction affecting the ability to produce high-pitched sounds, and modulating frequency and voice projection. There is a lack of consensus regarding the diagnostic criteria for EBSLN injury; however, laryngeal electroneuromyography (LEMG) is generally considered to be the gold standard examination [39]. Current studies on TLUSG did not aim to differentiate between VCP caused by RLN and EBSLN injury. This is likely due to a lack of pathognomonic features to diagnose EBSLN injury through laryngoscopic examination [40]. Although injury to EBSLN may lead to phase asymmetry, especially of arytenoids cartilage, which may be identified through ultrasound, the role of TLUSG to diagnose EBSLN-related VCP remains unclear [41]. Further studies are required to define ultrasonographic features of such injury and comparing the efficacy of TLUSG to LEMG for diagnosing EBSLN injury.

Previous authors have proposed potential algorithms for VCP screening using TLUSG. Knyazeva et al. suggested that confirmatory DFL assessment should follow TLUSG examination in patients with self-reported voice abnormality and in cases of reoperation procedures or patients with “nerves-at-risk” [19]. Through this review, we propose an algorithm to standardize TLUSG examinations’ protocol and effectively utilize various technical modifications to improve its diagnostic accuracy and VC visualization rates. (Figure 2) The algorithm also describes indications for mandatory laryngoscopic evaluation such as patients with un-assessable VC on TLUSG, diagnostic uncertainty with TLUSG, clinical urgency, and lack of experienced examiner or required equipment.

This systematic review and meta-analysis has limitations. The low incidence of VCP in included studies may contribute to low statistical power of our analysis and may have influenced the diagnostic accuracy of TLUSG. The differences in the study protocols of individual studies might have affected the performance of TLUSG. In the included studies, TLUSG was performed either in the immediate/early postoperative period (1–4 days) or in the late postoperative period (7–10 days). Different specialists with varying degree of experience performed TLUSG (surgeon, radiologists, anesthetists) and DFL/VL examination (surgeon, otolaryngologists, phoniatrics or endoscopists). There were also minor differences in the technical aspects of the examination, with some studies not performing the lateral approach and/or one of the three maneuvers (passive, active, valsalva). The diagnostic accuracy of TLUSG was better postoperatively than preoperatively, which may be due to paucity of preoperative data. Additionally, it may be due to the examiner identifying VCP symptoms in the former period. This presents a source of significant bias, which might be resolved to an extend by blinding performed by all included studies. On the other hand, it is clinically difficult to blind examiners to symptoms such as hoarseness as examiners must converse with the patients to collect required information and provide instruction during TLUSG examination. Finally, the criteria for diagnosing VCP through TLUSG was decreased (palsy) or absent (paralysis) movement of VC in the majority of the studies. This might have led to an overdiagnosis of VCP, which might be clinically irrelevant, therefore resulting in overestimation of TLUSG’s diagnostic performance.

## 5. Conclusions

In conclusion, TLUSG is a promising investigative modality, which through assimilative use of various technical modifications and increasing expertise has the potential for becoming the standard in a specific demographic (female and/or younger patients) for VCP diagnostics. Currently, it is an efficient screening tool saving 80% of patients from unnecessary invasive laryngoscopic examination. The inherent characteristics (non-invasive, quick, cheap, bedside/out-patient assessment) along with optimistic evidence demands further research into methods to elevate its position in the VCP diagnostic ladder. Future research must attempt to standardize and organize various modifications into an efficacious algorithm, develop and validate objective methods for assessing VC movement, and evaluate its learning curve and describe training protocols, in order to attain the optimal potential for TLUSG.

## Figures and Tables

**Figure 1 jcm-10-05393-f001:**
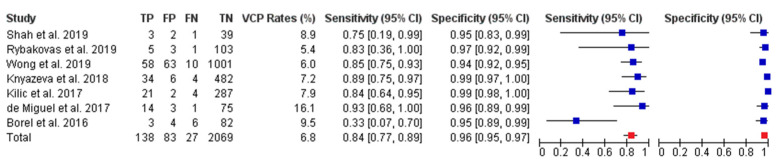
Forest plot of sensitivity and specificity of TLUSG examination in the postoperative period. Blue square indicates the summary estimates for individual studies. Red square depicts the overall summary estimate from all included studies. (TP = true positive, FP = false positive, FN = false negative, TN = true negative VCP = vocal cord palsy rate, CI = confidence interval).

**Figure 2 jcm-10-05393-f002:**
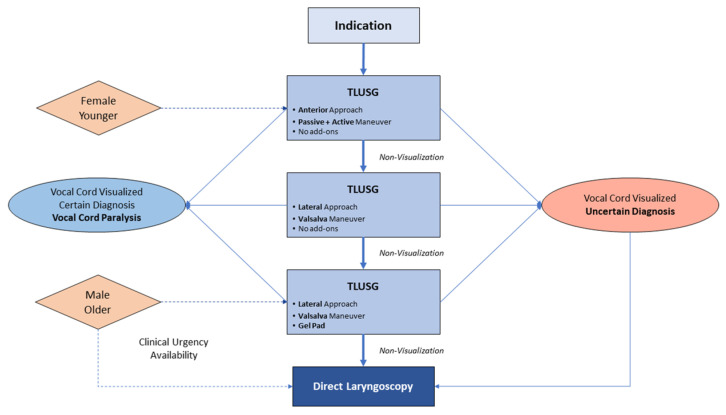
Algorithm to standardize TLUSG examination protocol through ordered use of technical modifications. It also describes considerations for compulsory laryngoscopic examination.

**Table 1 jcm-10-05393-t001:** Characteristics of studies assessing TLUSG in terms of operative period.

Study Characteristics				TLUSG Examination			Cohort Demographics		
**First Author, Year, Country**	**Accrual Years**	Study Design, Blinding	Reference Method	USG Probe, Frequency	TLUSG Approach	Maneuver Used	No. of Patients	Female%	Mean Age
Borel et al., 2016, France	2013–2015	P, Y	DFL	Linear, 8 MHz	Transverse Approach	A	103	82.5	51
de Miguel et al., 2017, Spain	2014–2015	P, Y	VL	Linear, 5–10 MHz	Transverse Approach → lateral approach (if non-visualization)	PA	108	78.5	NR
Gambardella et al., 2020, Italy	2018	P, Y	DFL	Linear, 7–13 MHz	Transverse Approach → lateral approach (if non-visualization)	PAV	396	66.2	56.4
Kandil et al., 2016, USA ^1^	2013–2014	P, N	DFL	Linear, 12 MHz	Transverse Approach	PA	250 (500 VC)	83	52.7
Kilic et al., 2017, Turkey	NR	P, NR	DFL	Linear, 6–13 MHz	Transverse and/or lateral approach	A	325	78.5	48.2
Knyazeva et al., 2018, Germany	2016–2017	P, Y	DFL	Linear, 5–10 MHz	Transverse Approach	PAV	668	83.8	50.3
Rybakovas et al., 2019, Lithuania	2016–2017	P, NR	DFL	Linear, 4–10 MHz	Transverse Approach → lateral approach (if non-visualization)	NA ^2^	112	82.1	56.2
Shah et al., 2019, India	NR	P, Y	VL	Linear, 5–10 MHz	Transverse Approach	NA ^2^	45	86.7	42
Wong et al., 2013, Hong Kong	NR	P, Y	DFL	Linear, 5–10 MHz	Transverse Approach → gel pad (if non-visualization)	PA	204	78.9	52 ^3^
Wong et al., 2015, Hong Kong	NR	P, Y	DFL	Linear, 5–10 MHz	Transverse Approach	PA	581	80.2	52 ^3^
Wong et al., 2019, Hong Kong	2012–2016	P, Y	DFL	Linear, 5–10 MHz	Transverse Approach	PA; PAV (from 2014)	1196	78.59	51 ^3^

(P = Prospective study, NR = not reported, Y = yes, N = no, DFL = direct flexible laryngoscopy, VL = video laryngoscopy, TLUSG = transcutaneous laryngeal ultrasonography, A = active phonation, P = passive breathing, V = Valsalva maneuver, NA = not applicable, VC = vocal cords) ^1^ Excluded from quantitative analysis due to extremely outlier results owing to diagnostic accuracy determination based on number of vocal cords instead of patients, and unreported visualization rate. The study is included in the qualitative analysis. ^2^ Not applicable since TLUSG examination was undertaken in the immediate postoperative period. ^3^ Median age.

**Table 2 jcm-10-05393-t002:** Weighted arithmetic means of diagnostic accuracy of TLUSG in the preoperative and postoperative period.

	*N*. of Studies	*N*. of Patients	VR% *	VCP% *	Sensitivity% ^§^	Specificity% ^§^	PPV% ^§^	NPV% ^§^
**Preoperative Period**	**3**	**1015**	**86.28**	**3.57**	**78.48 (NC)**	**98.28 (NC)**	**86.42 (NC)**	**99.73 (NC)**
Transverse + Lateral	2	489	95.93	6.93	91.15 (NC)	96.43 (NC)	71.82 (NC)	99.55 (NC)
Transverse alone	1	526	79.00	0.45	66.70 (11.85–98.29)	100 (99.40–100.00)	100 (17.77–100.00)	99.90 (98.93–99.99)
Valsalva Maneuver	2	922	85.50	3.61	79.67 (NC)	98.11 (NC)	85.05 (NC)	99.81 (NC)
No Valsalva Maneuver	1	93	94.00	3.20	66.70 (11.85–98.29)	100 (95.91–100.00)	100 (17.77–100.00)	98.90 (94.04–99.94)
**Postoperative Period**	**7**	**2317**	**94.13**	**7.11**	**83.95 (77.24–88.50)**	**96.15 (95.24–96.88)**	**64.32 (55.90–68.56)**	**98.71 (98.13–99.11)**
Transverse + Lateral	4	712	96.04	8.54	87.62 (78.16–94.33)	98.15 (96.81–98.94)	81.97 (70.85–89.28)	98.89 (97.78–99.47)
Transverse alone	4	1798	93.11	6.61	83.52 (74.52–88.16)	95.53 (94.44–96.42)	58.79 (49.20–63.81)	98.71 (98.03–99.15)
Valsalva Maneuver	2	1658	NA	6.39	86.63 (79.04–91.97)	95.56 (94.41–96.47)	59.67 (49.42–64.53)	99.06 (98.44–99.44)
No Valsalva Maneuver	4	1054	93.23	8.80	83.86 (75.08–89.97)	96.47 (95.10–97.46)	70.54 (60.59–77.39)	98.44 (97.39–99.03)
Radiologist	2	409	90.95%	8.27	72.15 (53.83–83.17)	98.39 (96.55–99.26)	80.06 (62.69–90.49)	97.36 (95.21–98.56)
Anesthesiologist	2	138	95.06%	0.82	87.33 (68.61–98.13)	95.84 (90.54–98.19)	75.09 (56.56–89.88)	98.31 (93.93–99.69)
Radiologist + Anesthesiologist	4	547	92.03%	7.00	75.98 (64.47–86.55)	97.75 (96.06–98.75)	78.81 (65.97–87.76)	97.59 (95.81–98.61)
Operating Surgeon	3	1770	95.06%	6.33	86.42 (79.07–91.71)	95.66 (94.57–96.54)	59.85 (49.86–64.61)	99.06 (98.46–99.43)

TLUSG = transcutaneous laryngeal ultrasonography, VR = vocal cord visualization rate, VCP = vocal cord paralysis rate, PPV = Positive predictive value, NPV = negative predictive value, NA = not applicable, NC = not calculated due to lack of data). One study was excluded from the entire quantitative analysis due to extremely outlier results owing to diagnostic accuracy determination based on number of vocal cords instead of patients, and unreported visualization rate. Two studies were excluded from postoperative analysis related to valsalva maneuver because relevant data was not available in the original study. For postoperative analysis, three studies reported on the same cohort, therefore, only the latest data for a particular analysis was considered. * Based on the total number of patients included in the entire study. ^§^ Based on the number of patients whose vocal cords were visualized on TLUSG examination.

**Table 3 jcm-10-05393-t003:** Risk Factors for non-visualization.

Author Name	Male Gender	Older Age	Body Mass Index	Disease	Thyroid Volume	Other Factors Examined
Kilic et al., 2017	+	+	−	−	−	Thyroid Function (−) Surgery type (−)
Borel et al., 2016	+	+	−	−	+	Postoperative drainage (+), Operative time (+) Cautery use (−), Reoperation (−)
Wong et al., 2015	+	+	−	−		Anatomical features ^1^ (−)
Gambardella et al., 2020	+	−	−	−		
Knyazeva et al., 2018	+	+				
Wong et al., 2013	+	+				
Kandil et al., 2016	−	−	+			Postoperative Period (+)

Red color (+) indicates that the factor is associated with vocal cord non-visualization risk, while green color (−) indicates no association. ^1^ Anatomical features include longer distance from hyoid to cricoid cartilage; shorter distance from cricoid to sternal notch; shorter distance from cricoid to incision.

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
