# Peer review of "Transcutaneous Laryngeal Ultrasound for Vocal Cord Paralysis Assessment in Patients Undergoing Thyroid and Parathyroid Surgery—A Systematic Review and Meta-Analysis"

_jcm, 2021, doi:10.3390/jcm10225393_

Round 1
Reviewer 1 Report
This report strengthens the data that supports the reliability of ultrasound to assess vocal cord mobility in thyroid and parathyroid surgery. In addition, it supports the technique of lateral and anterior placements of the probe along with Valsalva maneuvers. I do not detect any flaws or weaknesses in the methodology and conclusions. The authors are careful to indicated that this approach is for screening and not for definitive diagnosis of vocal cord pathology. The issue of superior laryngeal nerve injury should be discussed since ultrasound cannot detect this abnormality.
Author Response
Dear reviewer,
Thank you for reviewing our manuscript and providing your comments.
We have added a paragraph on external branch of SLN injury discussing its role in causing vocal cord paralysis and diagnostic issues relating to it.
Best regards,
Agastya Patel
Reviewer 2 Report
In the manuscript "Transcutaneous Laryngeal Ultrasound for Vocal Cord Paralysis Assessment in Patients Undergoing Thyroid and Parathyroid Surgery – A Systematic Review and Meta-analysis" the authors have chosen to assess a very interesting topic. The article as a whole is well written, although some minor points need improvement:
- In the abstract on line 13 you put the abbreviation TLUSG directly, but being the first time that you mention it, you have to put the entire sentence and then the abbreviation in brackets next to it.
- In lines 19 to 21 of the abstract show the various rates of visualization, sensitivity and specificity. Please, always put the confidence interval for each value between in brackets and specify which confidence interval you used for the calculation eg. 95%. Do the same for the rest of the manuscript.
- The introduction and Materials and Methods should be shortened as it is too long and cumbersome.
Author Response
Dear reviewer,
Thank you for reviewing our manuscript.
- Thank you for identifying our error. We have expanded the abbreviation of TLUSG in the abstract.
- We have added “CI 95%:” for those measures for which we were able to calculate the confidence intervals. We maintain the same throughout our manuscript.
- We have attempted to reduce the length of the introduction and methods section. In the methods section, there are certain pertinent points which must be reported in order to allow proper interpretation of our analyses.
Best regards,
Agastya Patel
Reviewer 3 Report
I am very grateful to the editor for including me as reviewer of this paper. It’s a very interesting work, which addresses a relevant issue in the clinical practice of endocrine surgery. The review and meta-analysis follow proper methods and highlights relevant points for the clinicians that employ or will employ Transcutaneous Laryngeal Ultrasonography (TLUSG) for the assessment of potential vocal cord paralysis. It fulfills the aim of the journal and is interesting for its readers. However, I would like to mention some points that could be improved.
- In the studies, different medical specialists perform TLUSG. An interesting analysis could be the comparison of diagnostic accuracy based on the specialist who perform TLUSG (radiologist vs others or radiologist vs anesthetist vs surgeon).
- I understand that this paper focuses on the recurrent laryngeal nerve (RLN) injury as an important complication following thyroid and parathyroid surgery. Nevertheless, superior laryngeal nerve can be impaired as well, especially in thyroid surgery, and this could lead to poor vocal quality. Although this lesion has less impact on patients’ quality of life than RLN lesion, it can be diagnosed with laryngoscopy. The ability of ultrasonography to detect this lesion could be commented in the Discussion.
Author Response
Dear reviewer,
Thank you for taking the time to review our manuscript.
- Thank you for your suggestion. We have added a subgroup analysis based on the specialist performing TLUSG. We present results for each kind of specialist: radiologists, anesthesiologists, radiologists + anesthesiologists and operating surgeon. The rationale behind such an analysis is that TLUSG has the possibility of becoming a point-of-care examination, which can be performed by the operating surgeons themselves, reducing the need for additional consultations for the patients. It appears that the diagnostic accuracy and vocal cord visualization rate in studies where the operating surgeon performed TLUSG examination is greater than the other specialists. However, this result must be considered in the light of the experience of the operating surgeon, which was greater than the other specialists.
- Thank you for pointing out the issue of SLN injury. We have added a paragraph discussing its role and related diagnostic issues in the discussion.
Best regards,
Agastya Patel